# Stein's Phenomenology of Grace

**Mette Lebech**

Department of Philosophy, Maynooth University, W23 X021 Maynooth, Ireland; mette.lebech@mu.ie

**Abstract:** Stein's *Freedom and Grace* (*Freiheit und Gnade*) phenomenologically describes the experience of grace as the desire, communication, or acceptance of God's Spirit of Love, accessed in the act of faith motivated by the soul's otherwise unfulfilled desire for self-mastery. This article first discusses the affordances of Stein's phenomenology which equip her to see grace as a fulfilment of the natural life of the soul, which is experienced as coming from beyond itself. It then addresses how the individual, personal I fails to satisfy its implicit desire for rational and free action in the natural life of the soul and how, in contrast, its opposite, the graced, liberated life of the soul, allows it to, but not on its own, only through union with God's Spirit. It proceeds from this existential alternative to show how the treatise unfolds as an investigation of the various *a priori* possibilities for grace to be experienced and why it makes sense to acknowledge faith as a legitimate source of knowledge, as Stein does in work postdating *Freedom and Grace*. Finally, it is argued that the treatise is phenomenological in nature and that it does not presuppose either metaphysics or Christian doctrine but instead contributes to underpinning both. This argument simultaneously explains Stein's own subsequent engagement as a Christian philosopher.

**Keywords:** Stein; grace; phenomenology; freedom; soul; theology; religion; faith; belief; Christianity

---

## 1. Introduction

Stein's Phenomenology of Grace constitutes her first attempt to account for the experience of divine love and may be read as the last of her early phenomenological writings.[1] Later, it would, as regards its focus on the experience, be followed up by her work on Dionysius the Areopagite and St John of the Cross. In both these instances, her analyses are conducted in relation to material from beyond the phenomenological tradition, whereas in this early treatise, she aims to look directly at freedom and grace in their relationship, studying them 'head on' as phenomena from within phenomenologically reduced experience.

The type of phenomenological analysis we find in *Freedom and Grace* is inspired by Husserl, Reinach, Scheler, and Conrad-Martius in equal measure. The analysis deploys both eidetic and constitutional analysis to the experience of grace as the communication of divine life towards the constitution of which faith contributes. The treatise thus proceeds without assuming any prior dogmatic foundation while being faithful to the phenomenological method alone. It identifies grace as superhuman motivatedness, or *Spirit*, issuing from the personal sphere or 'realm' of God, which can be intercepted and accepted as one's own through faith, whether this Spirit be felt or not, similarly to how values may motivate 'emptily' without the motivation being felt. Faith is described as an essentially freely performed act, which as a commitment occasions the handing over of the entire person to the life of the Spirit of God.

In Stein's work as a whole, *Freedom and Grace* plays the role of explaining how faith may be a legitimate source of knowledge and is requisite for reaching a view of the whole that is needed for metaphysics (Stein 2014, p. 99). Stein's defense of a Christian metaphysics in her educational anthropology (Stein 2004a, 2005c) and of a Christian philosophy in *Finite and Eternal Being* (Stein 2002a; 2006c, I, §4) ultimately relies on the phenomenological foundations laid in *Freedom and Grace*, since faith here is shown to be of a higher certainty

than any belief, however justified. The point of the treatise is not to lay this foundation, of course; rather, it is to phenomenologically explore the experience of grace and the act of faith that allows for it to be experienced—even if frequently only emptily—in its personal and intersubjective dimensions. It remains that it is by writing the treatise that Stein comes to understand how to proceed as a Christian philosopher.

In what follows, we shall first (1) look closer at the place occupied by *Freedom and Grace* in Stein's work as a whole and at the preceding phenomenological discoveries that allow her to envisage the experience of grace. Then, we shall (2) discuss the phenomenology of the contrast between the natural life of the soul and the liberated life of the soul that Stein proposes in the treatise. Next, (3) we shall give an overview of the whole of the treatise concerned to show how it identifies a priori ways in which grace can be communicated and discuss 'true faith' as the means to join the spiritual realm of God, from which grace flows in liberated life. We shall finally (4) discuss what is at stake in identifying the treatise as phenomenology and the reasons for doing so in light of recent literature on the matter.

## 2. The Role of *Freedom and Grace* in Stein's Work and the Phenomenological Discoveries Enabling Her Investigation of Grace

Stein's work is often divided into three periods. The works of the first period (1917–1921) are frequently characterised as early phenomenology and comprise *On the Problem of Empathy* (Stein 1917, 1989); the edition of Husserl's *Thing and Space* (Husserl 1973, 1997), *Phenomenology of Internal Time-Consciousness* (Husserl 1964, 2000), *Kantstudien* articles (Stein 2014) and *Ideas II* (Husserl 1989, 2002); the edition of Reinach's *Collected Works* (Stein 1921); *Contributions to the Philosophical Foundation of Philosophy of Psychology and the Humanities* (Stein 2000b, 2010a); *An Investigation Concerning the State* (Stein 2006a, 2006b); *Introduction to Philosophy* (Stein 2004b); and our treatise: *Freedom and Grace* (Stein 1962, 2009).

The works of the final period (1932–1942) are often qualified as pertaining to Christian Philosophy and comprise *The Structure of the Human Person* (Stein 2004a); *What is the Human Being?* (Stein 2005c); *Finite and Eternal Being* (Stein 2002a, 2006c, 2007c); translations of Pseudo-Dionysius and *Ways to Know God* (Stein 2000c, 2003b); *Science of the Cross* (Stein 2002c, 2003a); and various occasional writings, translations (Stein 2010b, 2013, 2020), and liturgical poetry (Stein 2007b).

The period in between these two, from *Freedom and Grace* to the commitment to Christian Philosophy (1922–1932), is possibly best understood as a transition. The works of the period comprise several translations: translations of Alexandre Koyré—*Descartes und die Scholastik* (Stein 2005a); of John Henry Newman—*Die Idee der Universität* (Stein 2004c) and *Briefe und Texte zur ersten Lebenshälfte* (Stein 2002b); and of Thomas Aquinas—*Über die Wahrheit* (Stein 2008a, 2008b). They also include five volumes of writings on phenomenology, ontology (Stein 2014), women (Stein 1996, 2000a), education (Stein 2001a), and the Christian life (Stein 2007a), notably 'What is Philosophy? A Conversation between Husserl and Aquinas' (1929) (Stein 2000c, 2014), Stein's first attempt at a systematic discussion of the relationship between Phenomenology and Scholasticism, and, following on from this lucid dialogue, her first large-scale sketch of an *Auseinandersetzung* between Modern and Medieval Philosophy: *Potency and Act* (Stein 2005b, 2009).

The fruit of this middle period is the recognition (1) that faith contributes to *experience as it is experienced* for the one who has it, (2) that the phenomenology of experience is foundational to philosophy, and (3) that Christian philosophy, i.e., philosophy founded on Christian experience, is therefore open to phenomenological investigation, if not in the first person (when this person does not have faith), then by empathetic insight into the experience of people who do have faith. If this triple recognition is the fruit and constitutes the rationale for doing Christian Philosophy, then the flower producing it is *Freedom and Grace*. In this treatise, we find Stein's reason for working from now on 'only in this area' of the philosophy of religion,[2] the thought process that led her in this new direction and made her embrace faith as a source of knowledge. The importance of the treatise for Stein's development therefore cannot be overstated: it is a phenomenological description of the



act of faith as it relates to its object: God in His goodness and graciousness, and of the transformation of the life of the soul which faith allows for as grace is experienced in its three modalities—longed for, communicated, and accepted.

Among the phenomenological discoveries that allow for Stein's investigation of grace, two must be singled out. They both rely on her signature investigation of psychic causality, complementary analyses to *Ideas II*, which, taken generally, is what allows her to make credible constitutional analyses of human reality (body, soul, and spirit).[3] These analyses allow for a constitution of the subject as psycho-physical (experiencing itself as 'having' both body and soul) and for a distinct experience of purely spiritual experience, identified as motivatedness. The counterpart to this purely spiritual experience is the experience of sentient or psychic causality, marking experience as containing a causal element manifesting in the degree and quality of aliveness experienced, ranging from excitement over vigour to tiredness and exhaustion. In the context of this article, we need to address Stein's understanding of the soul and of nature to understand how she can approach her subject.

To Stein, the *soul* is constituted (i.e., identified from within the stream of experience) as the substantial unity of the psyche, in turn constituted from all acts of an I in which *psychic causality* manifests (Stein 2000b; 2010a, treatise 1). The soul is therefore centred on the I and carried by the *person*, such that it is true to say, as we in general do, that we *have* a soul, like we have a body. The soul's *depth* is revealed in proportion to the height of the values to which the personality of the person is responsive, and therefore in extreme cases, it may remain dormant, for example, if a person is not expressing their own values but is dependent for its inner life on another person's value-response through sentient contagion. For this reason, the soul can be colonised, taken over, by the spirituality of someone else, and that can happen to a greater or lesser extent, in ways that may be conscious or not. It is this aptitude of the soul to be possessed, by myself or someone else, that takes centre stage in our treatise. The soul has this aptitude and vulnerability because of its causal nature: this is a capacity that also makes it radically accessible to God.

This brings us to our second feature: *nature*. The soul has *nature* because it is not pure spirit. It can experience itself to be owned, protected, moved, harmed, possessed, and redeemed because of its nature and by that which is pure spirit: *persons*, own or foreign, good, or bad. This possibility was gained insight into already in the constitutional analyses of *On the Problem of Empathy* and *Philosophy of Psychology and the Humanities*. Now, it is being treated directly in the experiential possibility of being possessed by someone else—as is foreshadowed in human love—and experiencing this possibly possessing spirit as capable of enabling me to do what I otherwise could not. Spirits, however, come in two kinds: those who want to possess to use and destroy, making a means of the one they possess to obtain power for themselves, and those who want to possess for the sake of the one they thereby redeem through their love. Distinguishing between them is an existential matter of ultimate importance, which we get to practice frequently in human relationships.

We might now see that the natural life of the soul is, because it is causal in nature, never so free that it cannot be had by something or someone other than the person whose soul it is. This leaves me constitutionally to yearn for the personal freedom to protect my soul's life, which is my substance and fullness, and without which my personal unity would be torn, and my being compromised, since I am a person of whom body and soul form integral parts. To be free, I must rise above my soul's life, embedded as it is in psychic causality, to hold and protect it. To do this, I need a spiritual strength that I only sometimes possess. My life, however, is not suspended in the moments where I do not have this strength: I am, therefore, finite, constitutionally vulnerable, and capable of surrender; the protection of my soul is not a foregone conclusion, despite it being a priori desirable.

It should be clear at this point that attention to the constellation of these phenomena—psyche, soul, psychic causality, sentient contagion, nature, person, and spirit—is characteristic of Stein's phenomenology and distinguishes it from Husserl's, Reinach's, Scheler's and Conrad-Martius'. This is not to say that these do not discuss these matters—they do, to different extents and in different combinations. It would be a study of its own to discuss the

extent to which each of these phenomenologists might confirm Stein's findings regarding the soul. None of them, at any rate, sees as clearly as Stein does the soul's constitutional need for a protection that I am only intermittently able to provide for it in myself, and none of them sees this need to manifest itself in the difficulty I experience in freeing myself from reactivity in my response to the world.

### 3. The Experienced Contrast between the Natural Life of the Soul and the Liberated Life of the Soul

Concerning the natural life of the soul, Stein says the following at the outset of our treatise:

> The natural, naïve life of the soul is a steady exchange of impressions and reactions. The soul receives impressions from outside, from the world, in which the subject of this life stands and which it takes spiritually as its object; it is moved through these impressions and through them, attitudes to the world are released in it: Fear or surprise, admiration or contempt, love or hatred, joy or grief. Willing and acting too. We referred to these as reactions and in the last examples—willing and acting—one usually speaks of activity. With a certain right, since the soul in all attitudes is in movement, in action, and by willing and acting this movement does not remain inside itself but rebounds back outside to impact on the external world. From a deeper standpoint however, one is justified in seeing this transfer mechanism of the natural attitudes as a *passive* one. And likewise, as *unfree*. For in all these movements there is no final initiation from a last inner centre. The soul subject is caught in them from the outside and have no control over itself in them. These two, however,—self-control and the initiation of one's movements—characterise activity and freedom in the strict sense. This passive activity, reaction as such, characterises the animal level of the life of the soul. (Hereby it is not excluded that certain such attitudes cannot be realised in the life of the soul of the animal.).

Stein continues:

> We contrast the natural, naïve life of the soul with one of an essentially different structure, which we (with an expression still in need of clarification) will call *liberated:* The life of the soul, which is not driven from the outside but *led from above*. The *from above* is simultaneously a *from within* since to be raised to the realm on High means for the soul to be set completely into itself. In that it is drawn into itself and thereby anchored in the Heights, it is also *pacified*, withdrawn from the defenseless exposure to the impressions of the world. Exactly that is what we refer to as 'liberated.'—The liberated soul-subject, like the natural naïve one, encounters the world with its spirit. It also receives impressions of the world in its soul. But the soul is not moved immediately through these. It takes them up to encounter them from exactly this centre through which it is anchored on High: its attitudes depart from this centre and are prescribed from on High. This is the soul-*habitus* of God's Children. Their freedom, the 'freedom of a Christian,' is not the freedom of which we spoke earlier. It is freedom from the world.

The kind of attitude-taking that corresponds to it is again a passive activity, but of a different kind than in the 'realm of nature.' The mechanism of the natural life of the soul does not touch its centre, which is the place of freedom and the point of origination of activity. The led soul obeys what is above in precisely this centre, conceives here the direction from above and lets itself be moved 'obediently' through it. The activity is bound in its place of origin: no use is made of freedom where freedom originates.

Stein is here making several points. First, the liberated soul encounters the world spiritually: it responds from on High, and it does not react directly through its causal nature to impacts from outside. Secondly, this freedom is not experienced as a freedom to choose my own attitude arbitrarily. Its responding is a responding to the things that occur in the world, and it makes sense in relation to these. They, moreover, command from on High a

response that is prescribed—by their value, first, and then by the value of the situation in its entirety, as it is seen from on High. The response keeps the soul at rest: the soul is not defenselessly impacted, nor does it act out. It is protected by the Spirit that responds, and by that Spirit's power, it has been set free from the world of causal (re)activity.

The attitude of responsivity does not cancel the causal nature of the soul. It is not making demands on the soul that would require it to have a different nature from the one it has. The new responsivity is a passive activity, one that does not have to either invent what to do (that is dictated through the values supported from on High) or provide the energy for doing it out of its own resources (since it is given through the motivating powers and conserved by the soul being taken up into the protective personal sphere emanating from on High). It obeys the direction given from above and relies on it for its freedom to rest in it. Through this freedom, the centre takes possession of the soul and pacifies it.

The description of the alternative between the natural and the liberated life of the soul is completed by descriptions of the autonomous and the possessed life, which also are alternative to it. What is thus portrayed is the experience of being faced with the existential choice of how best to protect one's soul. The experience of relating in acquiescence to the Above as the Above of the centre, through which one obeys in response to the world outside, is already one of faith, faith in the goodness of what comes from on High, which occasions the trust out of which the obedience makes sense.

In consequence, *Freedom and Grace* presents the experience of grace as a justification for religiosity (relationship with the Above) in general. It does so for Catholic Christianity as well, but to understand how, we need to look at the consequent development of the treatise.

## 4. An Overview of the Treatise as Identifying the a Priori Possible Ways in Which Grace Can Be Communicated and Therefore Relate to Freedom

The first section, from which we have so far quoted, is entitled 'Nature, Freedom, and Grace'. It shows how the soul gains fullness by joining a spiritual realm in which it can unfold—that of God or that of the devil. In both cases, the unfolding bears the stamp of the realm: in the first, it confirms and celebrates all that is; in the second case, it is *against* all that it is.

The second section concerns 'The Parts of Freedom and Grace in the Work of Salvation'. Having shown the necessity of joining a realm for the soul to have life (for its salvation), it is now shown that the soul cannot achieve this by its own powers or initiative: it must be taken up into this realm, to which it must acquiesce. It experiences this constitutional inability and its own collusion with it in anguish. From this experience, it may understand the need to turn away from itself and make itself available for being taken up. This may itself be experienced as prevenient grace and as such be accepted or rejected (rejected because the noticing of it will waken and heighten the feeling of one's own sinfulness). Yet, the possibility remains of throwing oneself without conditions into the arms of grace: this is 'the most determined turning away from itself, the most unconditional letting go' (p. 30).

In this endeavour, others can encourage by example and by attempting to bring grace and souls together, whether by teaching or by supplication on souls' behalf. This is 'The Possibility of a Mediating Salvific Activity', which gives the title to the third section. Everyone alive through grace can participate in this mediation and thus stand for all before God. Through sinfulness and lack of charity, however, this mediation is ineffective without the mediation of God's own Christ, who graciously makes good any lack of initiative and perseverance that lesser mediators may fall prey to.

The fourth section, 'The Psycho-Physical Organisation as Point of Engagement for Salvific Activity,' shows how it is possible that grace may reach the soul through the body in sacraments because of the unity of body and soul in the human being as it lives in the world. Whether there are such sacraments, and what they are, are determined only through faith, which if obtained, in turn embraces grace through them. Therefore, the last and fifth section is dedicated to an exploration of 'Faith' and its relationship with similar acts such as belief.

According to our treatise, faith is the act by means of which I may join the spiritual realm of God and through the centre of my soul give myself over to its spiritual direction and protection and live with its life in the world. The communication of grace can happen in five ways, each of which is treated in a section of the treatise. Faith plays a central role in each but is articulated differently and involves different beliefs in each case. Grace can be communicated directly (1) as the soul through its centre obeys directions from on High and responds to the happenings in the world. In this case, faith is exercised only as assent to this direction and may be experienced as contentless apart from it. This assent is nevertheless the essential ingredient in all the other forms faith may take. Grace can be gained insight into (2) as indispensable for retaining freedom in the face of the world and obtaining liberation or salvation from it. In this case, faith takes the form of trusting grace's ability to provide this freedom. Grace can become present to us through others (3), and we may bring it close to them in their or our acceptance of it and in communication about it. Faith, in this case, incorporating the features already mentioned, involves awareness and identification of holiness as it is lived out, as well as of the characteristic features of this life. Faith is placed not only in the Above, but is also seen as invested in human beings as they live a holy life, intercede for us, and mediate our relationship with God. Faith may even focus on one such, if, as in the case of Christ, the mediation of that person is believed to be indispensable. In this case, faith extends itself to all that this person is and to the Church as the assembly of the faithful representatives of this person. Grace can be handed to us through its sacraments (4), i.e., through communication involving the body, since the body allows for the soul to be affected through it. Belief in this being the case and in the importance of the sacramental effect may extend faith to the sacraments, such that faith is lived through a sacramental life. Finally, faith may stand on its own as an act (5) in which all or some of the above forms are sanctioned, involving an entire worldview, possibly inherited through the faith of previous generations, but active only if *I* believe.

In this way, *Freedom and Grace* describes the experience of grace as longed for, communicated, rejected, and accepted, but concentrates on the experience involving acceptance, since its clearer view of it testifies to its author having personal experience of it. Thus, *Freedom and Grace* describes 'religious' experience simultaneously as characteristic of someone who *has* faith and as the set of reasons why someone *would want to have* faith because faith is 'called for'. It seems plausible that this faith would be accompanied by a set of beliefs articulating this faith and that it would in consequence be understood to be faith in God, in Christ, and in the Church. However, the set of beliefs relies on that which gives faith its substance: the existential acceptance of grace. If beliefs are detached from faith, they will constitute a worldview among other worldviews. Faith, on the other hand, might possibly be had without any such worldviews, although this may seem comfortable only in situations of extreme doubt and may in those cases coexist with agnosticism.

## 5. Phenomenology or Not Phenomenology, That Is the Question

It may sound strange to claim that it constitutes an a priori possibility for grace to relate to freedom through others and through the psycho-physical organization of the body. This may well be the reason why, contrary to what we claim to show here, our treatise hitherto has only rarely been seen as phenomenology and in consequence to pertain to Stein's early phenomenological writings.

Until Mariéle Wulf's 'Rekonstruktion und Neudatierung einiger früher Werke Edith Steins' from 2003[4] established the work's correct title as *Freedom and Grace* and convincingly proposed the date of 1921 for its completion instead of the hitherto assumed 1932, the work was understood to belong to Stein's later, explicitly Christian, philosophy. Beate Beckmann treated it as such in her *Phänomenologie des religiösen Erlebnisses* (Beckmann 2003) and therefore read it as a work presupposing metaphysics and Christian dogmatics.

Nicolas Vinot Préfontaine used the work extensively in his Salzburg dissertation from 2008: *Metaphysik der Innerlichkeit. Die Innere Einheit des Menschen nach der Philosophie Edith Steins* (Préfontaine 2008). But, like Beckmann, he did not refer to it as phenomenology. His

interest was in the *metaphysics* of the inner person and its contribution to the understanding of the normative unity of the human being. An anchoring of such metaphysics in the a priori explorations of the I and the person characteristic of Stein's early work, including *Freiheit und Gnade*, would, however have strengthened his argument considerably, as it would have Beckmann's to read *Freedom and Grace* as a response to Reinach's *Aufzeichnungen*. We might therefore presume that it did not occur to either to read the treatise as phenomenology, possibly because classical phenomenology was thought of as incapable of dealing with such phenomena, or because it felt wrong to regard grace as a phenomenon. This, however, is an expectation we need to abandon if my reading of the treatise is correct. Then, we must give it a hearing that grace is a phenomenon that Stein attempted to submit to a phenomenological investigation, and we can begin to discuss this investigation's plausibility and importance for both theology and philosophy.

When the work was published in ESGA 9 in 2014 (Stein 2014) under the correct title and in the correct chronological context, the editors Beate Beckmann and Hans Rainer Sepp are of two minds as to whether to describe the work as phenomenology. They claim on the one hand that 'Stein conducts no correlation-research involving the connection between noema and noesis in human consciousness' (p. xxx)[5] and that she approaches her topic 'without the attitude of transcendental phenomenology typically present in her other works.' (p. xxix). On the other hand, they see 'her own correlation-research results from her early works [ . . . to be . . . ] partly retained,' (pp. xxix–xxx) and classify the work as 'philosophical *phenomenology* of religion' (my emphasis), although that is quickly qualified as 'psychology of religion on a phenomenologico-philosophical basis'. If my above analysis is correct, *Freedom and Grace* is unambiguously phenomenology, as the treatise *describes* the experience of grace desired, offered, accepted, and rejected, as well as the acts in which it is being experienced according to these modalities. It is indeed 'correlation-research', in that it investigates both the noematic and noetic aspects of the experience of grace, as well as the role played by freedom in determining the experience of it. What may cause confusion is Stein's focus on the *natural* and *supernatural* life of the *soul*—the psycho-spiritual unity of a *human person*—since it is easily forgotten that Stein contributes towards clarifying the soul's constitution along with that of the human person. We pointed this out in our first section as one of Stein's phenomenological discoveries: these are referred to by Beckman and Sepp as 'correlation research results.' We also pointed to another such result concerning nature. The term 'natural' may in fact also form an obstacle, although the constitution of nature through the lawfulness of causality that regulates it is commonplace in classical phenomenology. If one neglects these discoveries or correlation-research results reinforced by her analysis of psychic causality and its relationship with motivation, it becomes easy to mistake them in the treatise for testimonies to a metaphysical position. We should not, however, do that, since phenomenology can contribute significantly to contemporary and future fundamental theology in a way that metaphysical positions cannot: it gives experientially first or ultimate reasons, not reasons founded on anything less, such as metaphysical construction or even Christian dogma.

Christof Betschart's edited collection, *La liberte chez Edith Stein* (Betschart 2014), highlights the phenomenological approach. Sr Cécile de Jésus-Alliance (Rastoin) ocd beautifully characterises phenomenology as 'this step-by-step philosophy, this maieuticism of Modernity [which] is particularly able to guide our sojourning *en toute liberté*' (p. 7). Fr Denis Chardonnens ocd's '*Liberté et grâce*. La contribution d'Edith Stein à une réflexion d'anthropologie théologique' (pp. 137–60) follows this up without attempting to assess the nature of the work, however. Since he understands her contribution as a contribution *to* theology, without classifying it *as* theology, he is nevertheless able to profit from it *for* theology because he intuitively understands it to rely on experience.

Tonke Dennebaum is, in his *Freiheit, Glaube, Gemeinschaft. Theologische Leitlinien der Christlichen Philosophie Edith Steins* (Dennebaum 2018), looking for the 'theological content' of the treatise. In this, he may or may not see that the phenomenology of grace is what *provides this content to* theology without it itself being theology, i.e., without it relying on the

Revelation the subjective reception of which it describes. The same reluctance to identify the treatise as phenomenology can be found in Laurence Bur's otherwise excellent thesis *La liberté selon Édith Stein: un chemin entre deux abîmes* (Bur 2021). Unwilling to engage the treatise as phenomenology, it is classified chronologically according to Gelber's earlier hypothesis, with the result that the transition to Christian philosophy is left without an essential, explanatory element.

Although Stein's turn to a Christian philosophy may mark a turn to meta-phenomenological position, it does, as Mariéle Wulf insists in a recent publication (Wulf 2018), make sense that Stein would remain a methodological phenomenologist throughout all her works, since she at no point explicitly rejects this methodology and consistently praises its capacity to contribute to the clarifying of concepts. To Ingarden's proposal to discuss her earlier phenomenological works while she was composing *Freedom and Grace,* she does answer that these are to her now 'like the shed skin of a snake might be to it'. But she also states that there is nothing in them 'objectively' over which she would want to go back (22/9-21 ESGA 4, p. 142).[6] *Freedom and Grace* builds on these treatises, although it also surpasses them towards something new. It is, however, not yet characterised by that meta-phenomenological position which it labours to define and for which it provides the rationale, but it clearly constitutes the path towards it, as well as the key to understanding it.

## 6. Conclusions

I regard—generally speaking—the phenomenology of grace presented in Stein's 'Freedom and Grace' as plausible, not to say overwhelmingly credible and enlightening. I have not, however, attempted here to discuss it *as* phenomenology, i.e., to criticize it in the light of the experience itself. I have merely attempted to provide an argument for the fact that we must, to understand both the treatise and the role it plays in Stein's development, read it in this way, *as* phenomenology.

Phenomenology may mean different things to different people. Stein may also over her career have adjusted her understanding of it several times. Let us therefore sum up our discussion in relation to a quote from Husserl's fifth *Cartesian Meditation* (Husserl 1960, p. 150), where he clarifies what phenomenological explication is in relation to what it is not and with which Stein arguably remained in agreement all her life:

> [ . . . ] phenomenological explication is nothing like 'metaphysical construction'; and it is neither overtly nor covertly a theorizing with adopted presuppositions or helpful thoughts drawn from the historical metaphysical tradition. It stands in sharpest contrast to all that because it proceeds within the limits of pure 'intuition', or rather of pure sense-explication based on a fulfilling givenness of the sense itself. Particularly in the case of the Objective world of realities (as well as in the case of each of the many ideal Objective world, which are the fields of purely a priori sciences)—and this cannot be emphasized often enough—phenomenological explication does nothing but *explicate the sense this world has for us all, prior to any philosophizing,* and obviously gets solely from our experience—*a sense which philosophy can uncover but never alter,* and which, because of an essential necessity, not because of our weakness, entails (in the case of any actual experience) horizons that need fundamental clarification.

Grace has clearly in *Freedom and Grace* been investigated according to its sense: in relation to what it means when longed for, communicated, rejected, and accepted. No helpful thoughts or adopted presuppositions have been appealed to, although many would confirm it. Admittedly, grace might not belong to the 'Objective world of realities' for all, in the sense that only some seem to have intimate, personal knowledge of it from having accepted it. However, this is not specific to grace: coyotes and racoons also are animals of which not everyone has firsthand knowledge. They, however, like grace, belong to the sense this world has for all of us, and which it obviously obtains from our experience, which we can communicate with each other and thus share. Philosophy can uncover this sense but cannot alter it. The sense of grace and its relation to freedom is what Stein's treatise has

brought closer to us, whether or not we have first-hand experience of it. She has brought it closer—but not closed the discussion about it—since it, according to its essence, like coyotes and racoons according to theirs, entails infinite horizons in need of fundamental clarification.

What Stein has not done in the treatise, however, is propose a reflection based on adopted presuppositions or helpful thoughts drawn from tradition. Nor has she proposed a metaphysical construction or called on a naïve faith for which no account can be given. For all of these reasons, theology can profit from this work more than it can from any already theological reflection: it can draw from the experience itself, which it in fact already relies upon without clear awareness of the methodology that can bring it to full and pure clarity. Here's to theology taking account of phenomenology as the discipline that also can help it—like it can help all the sciences—to identify its object more clearly!

**Funding:** This research received no funding.

**Institutional Review Board Statement:** Not applicable.

**Informed Consent Statement:** Not applicable.

**Data Availability Statement:** Not applicable.

**Conflicts of Interest:** The author declares no conflict of interest.

## Notes

[1] The material for this article was first presented at the conference *The Future of Christian Thinking* in Maynooth April 2022.

[2] Letter to Ingarden, 30 August 1921 (Stein 2001b, 140; CWES 12, 188). The treatise referred to was, according to the same letter, written after she finished *An Investigation Concerning the State* and is thus the last work Stein wrote before her baptism on New Year's Day 1922.

[3] *Philosophy of Psychology and the Humanities I: Psychic Causality* (Stein 2000b; 2006a; 2006b, 9-128; 2010a, 5-109).

[4] It was published in an important collection by (Beckmann and Gerl-Falkovitz 2003, pp. 249–67).

[5] All translations from ESGA are my own.

[6] 'Sachlich habe ich kaum etwas daran durchzustreichen.'

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
