# Peer review of "Stein’s Phenomenology of Grace"

_religions, doi:10.3390/rel14070950_

Round 1
Reviewer 1 Report
The article has many qualities: it is well written and the thematic is interesting and relevant for this special issue. The author is clearly well oriented in Stein’s authorship and in the current research literature, and s/he makes several valuable observations. So, in the following, as is the duty of a referee, let me focus on what may be improved.
My main objections to this as an article are that (1) the question, the issue investigated, is not clearly formulated and consequently (2) there is a a lack of balance and coherence / wholeness in the disposition and argumentation of the text.
So: What is the question at stake? We would suggest the author clarifies this and then relate each step in the development of the exposition more explicitly to his/her chosen overarching question.
Let me quote some possible issues as formulated in the article. The project is to explore and / or argue for: – the character / contribution of FG? – defend the position that FG is unambiguously phenomenology? – the rationale for doing Christian Philosophy? – faith as a legitimate source of knowledge? – that the experience of grace is an explanation for, and indeed a source of, Christian doctrine? – or other?
I shall try to exemplify the above by looking at the article section by section.
[line 17sq] Introduction
The author is placing FG (Freiheit und Gnade) in Steins work. Then s/he describes the context and background of the writing of the FG-treatise and gives a rather detailed account of how the manuscript for a long time was assigned an inaccurate name and date. The account is important for the thematic of the present article, but as by now the story is well known and extensively accounted for both in ESGA vol. 9 and elsewhere, it could preferably be shortened. Also, the descriptions of what / who probably has influenced Stein during this particular period is relevant but again somewhat long.
The characterization of FG and its place in Stein’s work [line 70-104], gives a good and succinct summary of the phenomenological contribution in this treatise and is well directed towards the aim(s) of the present article.
[line 105sq] “The role of Freedom and Grace in Stein’s work and current research on it “
The different periods in Stein’s work are well accounted for, leading up to a pertinent interpretation of Steins stand at the time of FG and the importance of this treatise in her development [line 130-140].
But then, after a return to the question of the dating, the remaining part of this long section consists in extensively discussing various commentaries in the reception of FG, concerning their interpretation of it as phenomenology or not (... or rather metaphysics or meta-phenomenology ...). The author is well acquainted with the research literature here, but again one wonders; is this the problem investigated, is this the central issue specified for this article? It is not clear.
Also, one might question the authors interpretations of the research contributions referred to, hinging on an implicit, not clarified, definition of ‘phenomenology’. Is there for example a dichotomy, an either-or as it seems to be proposed here, or are there variations in what may count as phenomenology, its boundaries being more fluid (Spiegelberg)? Neither the “either” nor the “or” is specified here, and one would expect the many discussions of the question “what is phenomenology”, from Ricœur to todays “French” debates, to be at least mentioned.
Only after discussing at length the various views on FG and phenomenology, starts an exposition of the work itself.
[line 222sq] “The phenomenology of the natural life of the soul and the affordances of Stein’s version of phenomenology allowing for it” and [line 289sq] “The liberated life of the soul and an overview of the treatise”
The author comments on a reading of FG, section I.1 [line 222sq] and section I.2 [line 289sq] in pertinent and well conducted analysis. Related to our concern above – what is the main question – one would wish the commentaries to be more explicitly linked to the overall interrogation of the article. The role of these analyses in the argument of the article is still not clear: is it f. ex. to show “the characteristic of Stein’s phenomenology” that distinguishes it from others [line 279]? or the coming to be of an experience of faith [line 333]? And then the notion of ‘grace’ is introduced all of a sudden [line 334] ... ...
Rather than in the following paragraphs just enumerating sections of FG, it would be more interesting to focus and elaborate in detail on this notion of ‘grace’ with references to Edith Steins own text, as the notion, judging from the title of the present article, is most central. This would also provide the substance of what the next section “In conclusion” seem to be summing up.
[line 369sq] In conclusion: ‘True faith’
Here, finally, the notions of ‘faith’ and ‘grace’ are addressed directly, and the summary is excellent! But it would need a foregoing exposition with detailed references to the text of FG itself as evidence for the suggested interpretations.
Rather abruptly, this very short section is called a “conclusion”, and yes, it does in fact look like a summing up. But the impression so far is that the argument has hardly started! As said above, we need a further elaboration of the arguments based on explicating Stein’s own text.
We would suggest that this section be elaborated as a section apart, before a final section as a Conclusion draws all the threads together in a more precise answer to a question posed.
And also; I cannot see “Divine Love” mentioned anywhere? Titles oblige ...
Two formal points:
– The long citations from FG, ESGA vol. 9, p. 10-11 and p.11-12, should imperatively be indicated as a citation which is not done (present article, line 224-241 and line 291-309).
– In the reference list it would be helpful to have the ESGA volume numbers indicated for each of the Stein-titles (line 408 sq) which is not done either.
Thank you for the read and good luck with the final finish!
Reviewer 2 Report
This article shows an excellent knowledge of Edith Stein's analysis of faith in Freedom and Grace. Nonetheless, this paper is more a thorough review of Stein's text than an actual research article engaging with the topic and arguments developed in the book and with the second literature on the issue. So the paper should either be submitted as a "review" or should be rewritten and reformatted to meet the standards of the Journal.
Round 2
Reviewer 1 Report
Thank you for a very well conducted re-writing of your article, making of it a solid contribution to our field!
Author Response
I shall keep in mind for the final reading whether I can still further clarify my research questions.
Reviewer 2 Report
The paper has been improved. Perhaps the final section and the reference to Husserl's definition of phenomenology could be revised. References to the Idea of phenomenology, phenomenology as a rigorous science or Ideas II (that Stein partly wrote and edited) would strengthen your point and be more relevant.
Author Response
Thank you for your comments. I shall keep in mind for the final review to still further clarify research questions and explain how results are obtained.
I could chose passages similar in content from the works proposed. I do not think it would improve the quality of the paper or strengthen the argument, however. The reason is the following: Those who might consider Stein an unaccomplished phenomenologist are to a significant extent those who think Husserl was justified in his turn towards transcendental idealism, because they consider like Husserl (and perhaps justifiedly so) that phenomenology as such implies such an idealism (in the sense of a self-contained sense of the apriori). That Stein could possibly agree with this, and that her phenomenology of grace also could is thus part of the underlying message sent in chosing the text I did. I shall keep in mind for the final revision how I might render this a little more explicit without entering upon a further argument that would go beyond the limits of the article.